

# Knowledge of cardiovascular disease risk factors among caregivers of cardiology patients attending Jordan University Hospital

Hanna Al-Makhamreh[1,*], Amro Alkhatib[2,*], Ahmed Attarri[2], Ahmad A. Toubasi[2], Aya Dabbas[3], Basel Al-Bkoor[2], Zaid Sarhan[2] and Osama Alghafri[2]

[1] Cardiology Division, University of Jordan, Amman, Jordan
[2] School of Medicine, University of Jordan, Amman, Jordan
[3] Faculty of Medicine, Al-Balqa Applied University, Al-Salt, Jordan
* These authors contributed equally to this work.

Corresponding author
Amro Alkhatib,
amralkhatib8@gmail.com

## ABSTRACT

Cardiovascular disease (CVD) is an umbrella term that includes various pathologies involving the heart and the vasculature system of the body. CVD is the leading cause of death worldwide, accounting for an estimated 32% of all deaths. More than 40% of annual deaths in Jordan are due to CVD; this number is further expected to rise, particularly in the Eastern Mediterranean region where Jordan is located. Due to the chronic nature of CVD, the presence of a caregiver who can help mitigate the challenges patients face is essential, and their level of knowledge determines the quality of care they can provide. Hence, this cross-sectional study was conducted in the cardiology clinics at Jordan University Hospital (JUH). Questionnaires were distributed to 469 participants, defined in this study as the caregivers escorting patients with established coronary heart disease (CHD). The self-administered questionnaire included three sections: sociodemographic and health factors, knowledge of CVD risk factors, and CHD symptoms. The mean age of the study population was 44.38 years ± 15.92 and 54.2% of participants were males. Regarding knowledge of CVD risk factors, 84.6% of participants answered more than 70% of the questions correctly. More than 95% knew that chest pain is a symptom of an acute cardiovascular event. However, only 53.5% and 74.8% of the participants reported that jaw pain and arm pain are symptoms of an acute event, respectively. Several factors influenced the caregiver's knowledge, such as age, income, frequent health checkups, having a history of CVD, CKD, or DM, and their relationship to the patient. This study sheds light on the importance of caregiver knowledge in patient care. By improving the caregivers' knowledge, identifying their role in patient care, and raising CVD awareness in susceptible populations, healthcare professionals can improve the patients' quality of life. Overall, assessing caregivers' knowledge pertaining to CVD can provide invaluable data, which may enhance patient care by educating their caregivers.

## INTRODUCTION

Cardiovascular disease (CVD) is a broad term encompassing pathologies of the heart and blood vessels, including those of the brain (*Okour et al., 2019*). CVD is one of the leading causes of death globally, accounting for an estimated 32% of all deaths worldwide and more than 40% of deaths in Jordan annually (*World Health Organization, 2021*; *Mukattash et al., 2012*). CVD rates in Jordan will soon rise significantly due to Jordan's fast socioeconomic development and the adoption of detrimental lifestyle habits such as unhealthy diets, physical inactivity, and smoking, all of which are becoming more prevalent in Jordan and are considered the drivers of the region's rapidly growing CVD burden (*Al-Nsour et al., 2012*).

Coronary artery disease (CAD), which falls under CVDs, occurs when the coronary arteries supplying the heart are narrowed, mainly by atherosclerosis. Coronary heart disease (CHD), on the other hand, includes the diagnoses of stable angina and acute coronary syndrome (*e.g.*, ST elevation myocardial infarction, non-ST elevation myocardial infarction, and unstable angina), all of which are associated with a high mortality (*Ralapanawa & Sivakanesan, 2021*). CAD is the third leading cause of death worldwide and is associated with 17.8 million annual deaths. In 2001, The Global Burden of Disease (GBD) estimated that 43% of all CVD deaths are related to CAD (*Ralapanawa & Sivakanesan, 2021*). In Jordan, CHD accounts for 54.7% of annual deaths (*Raffee et al., 2020*).

Due to the modifiable nature of most CVD risk factors, the development and progression of CVD is greatly preventable (*Schenck-Gustafsson, 2009*; *Pearson et al., 2002*). Knowledge of CVD risk factors is a prerequisite for preventing and controlling the disease (*Brown et al., 2005*). Hence, promoting a healthy lifestyle and educating the populace on the signs and symptoms of CVD can aid in the prevention and early recognition of cases while simultaneously improving the outcomes for those already diagnosed with the disease (*Becker et al., 1977*; *Ford & Jones, 1991*; *Yoon et al., 2001*).

A caregiver is an unpaid person who provides care to someone in need of assistance because of an acute or chronic condition; they could be a family member, a friend, or a neighbor (*Reinhard et al., 2008*). Caregivers are the foundation of long-term care for those suffering from physical and mental illnesses (*Parmar et al., 2022*). Their involvement helps patients better manage their condition and may prevent medical errors (*Mitnick, Leffler & Hood, 2010*). To the best of our knowledge, no published research to date thoroughly examined the knowledge of CVD risk factors and CHD symptomology among caregivers of CVD patients in Jordan. This cross-sectional study aims to assess caregivers' knowledge and current level of understanding regarding CVD risk factors and CHD symptoms while also examining the factors that may influence their knowledge of the topic. Caregivers of CVD patients attending a tertiary-level cardiovascular hospital in Jordan were included in this study. This study will help highlight the gaps of knowledge present in the caregiver population, which can aid in directing educational initiatives and counselling services to fill such gaps. In turn, enhancing caregiver knowledge will help them take better care of their health and their patients'.

## MATERIALS AND METHODS

### Study design and ethical approval

This cross-sectional study was conducted between March 2023 and April 2023 in the cardiology outpatient clinics at Jordan University Hospital (JUH). Ethical approval was granted by the institutional review board, Jordan University Hospital, University of Jordan (reference no. 10202310504). Each participant provided oral and written consent after the researchers informed them of the nature of the study and the objectives of the questionnaire. Researchers also informed the participants that the collected data was anonymous and only group-level (not individual) findings will be reported. The data remained confidential at all times, with only the research team members accessing it.

### Participants

Researchers approached caregivers from both genders who were at least 18 years of age and were escorting patients with an established CHD for an outpatient clinic appointment. The researchers introduced themselves, confirmed that the individual met the aforementioned definition of a caregiver (which was established by *Reinhard et al. (2008)*), and then proceeded to explain the nature of the study to the participants. If a patient came in with more than one caregiver, the caregiver who spent the most time providing care to the patient was asked to participate in the study, thus we included 1 caregiver for each patient. The calculated sample size was 365 to achieve a 95% confidence level ($P$-value $< 0.050$) for a population size of three million. An extra 25% of the calculated sample size was approached to account for the refusal of some individuals to participate in the study. Hence, the researchers approached 500 caregivers, 10 of whom did not provide informed consent, while 21 refused to participate due to time constraints. Thus, the total number of participants included in the analysis was 469 caregivers.

### Data collection

The researchers administered a questionnaire to the caregivers as a printed-out document. For some participants who could not read the questionnaire, the researcher filled out the questionnaire for them depending on their answers. Data collection occurred on all days of the week (except weekends and public holidays). Participants took approximately 10 min each to complete the questionnaire.

### Instruments

The self-administered survey was created using validated questions from an extensive literature review and translated back-to-back into the Arabic language (*Pallangyo et al., 2020*; *Swanoski et al., 2012*; *Aziz et al., 2008*; *Crouch & Wilson, 2011*). However, its distribution was in both the Arabic and English languages, depending on the participant's preference.

### Survey parts

The questionnaire consisted of three sections, which examined the sociodemographic factors, awareness of CVD risk factors, and knowledge of CHD symptoms.

- **First section:** addressed sociodemographic and health factors of subjects including their age, gender, education, marital status, occupation, income, residence, relationship to patient, personal medical history, family history of CVD, smoking status, physical activity, weight, and height. This helped us characterize the participants and account for possible factors associated with CVD knowledge.
- **Second section:** studied the participants' awareness and understanding of CVD risk factors and its questions were close-ended with a yes/no format. The risk factors examined were smoking, diabetes, high blood pressure, alcohol drinking, cholesterol, obesity, diet, physical activity, stress, and family history. There were 18 questions in total, which were gathered from an extensive literature search (*Pallangyo et al., 2020*; *Swanoski et al., 2012*; *Aziz et al., 2008*; *Crouch & Wilson, 2011*). Answers were then divided into correct/incorrect and a score was calculated for each participant by dividing the sum of correct answers by the total number of questions. A score of <50% was classified as low, 50–69% was moderate, and ≥70% was good knowledge, based on already validated cut-off points from the literature (*Pallangyo et al., 2020*). This section aimed to assess caregivers' knowledge about CVD risk factors, which helped us identify gaps in their knowledge that may later help tailor interventions.
- **Third section:** studied participants' knowledge of CHD symptoms. We asked if they identified the following symptoms as manifestations of CHD: chest pain, jaw pain, arm pain, fatigue, and shortness of breath. This section intended to assess caregivers' knowledge about CHD symptomology, which helped us identify gaps in their knowledge that may aid in determining interventions later.

### Data analysis

All data were exported into Excel sheets before being imported into the Statistical Package for Social Sciences (SPSS) version 25, which was used for all statistical analyses. Percentages and counts were used to present categorical variables; mean and standard deviation were used to interpret continuous variables. To assess the association between the variables and participants' knowledge, the Chi-square test was conducted. Any $P$-value < 0.050 was considered statistically significant.

## RESULTS

### Characteristics of the included participants

The study population consisted of 469 caregivers and had a mean age of 44.38 years ± 15.92. The males were 54.2% of the study population and the females were 45.8%. Most participants lived in urban areas (80.6%) and 63.8% went through undergraduate education. Furthermore, 47.3% of participants had income of less than 500 Jordanian dinars per month. Most caregivers were the children of patients 44.1%, followed by spouses, siblings, parents, and friends. Moreover, 55.7% of participants rated their general health as good and 55.0% had a health check within the last year. Regarding comorbidities

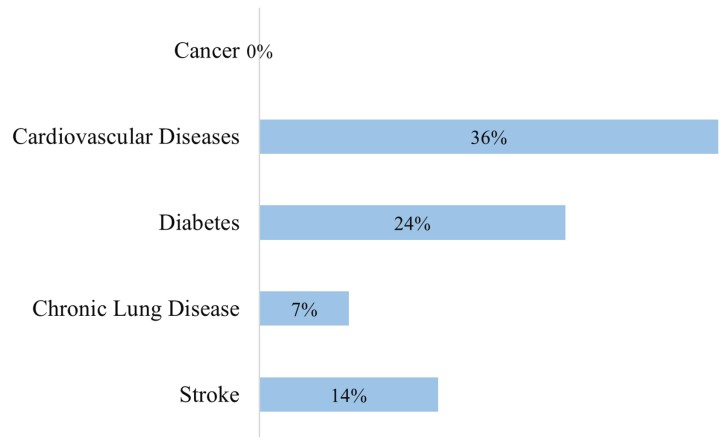

**Figure 1  Distribution of comorbidities among the participants.**

(Fig. 1), 36.5% and 23.9% of the patients had CVDs and diabetes mellitus, respectively. Only 10.4% of participants were current smokers, 42.1% were overweight, and 24.0% were obese. Characteristics of the included participants are demonstrated in Table 1.

## Caregivers knowledge about cardiovascular diseases

Analysis of the overall knowledge of participants revealed that 84.6% of them had good knowledge, which was defined as answering more than 70% of questions correctly (Fig. 2). Only 46.5% of participants reported that CVDs are chronic diseases, (67.6%) knew that CVDs are the main cause of death worldwide, and 84.9% knew smoking is a major contributor to CVDs. On the other hand, 80.8% and 95.3% of participants knew that diabetes and hypertension increase CVD risk, respectively. Additionally, 85.9% of participants reported that hyperlipidemia increases CVD risk, and 82.9% knew aging is a risk factor for CVDs. Furthermore, 69.9% of participants knew men have a higher risk for CVD, and 71.4% of them reported that CVDs are preventable. In addition, 71.2%, 72.5%, and 78.5% of participants knew LDL is bad cholesterol, blood pressure higher than 140 is considered hypertension, and passive smoking is harmful, respectively. Percentage of correct response to each question is demonstrated in Table 2.

## Knowledge about the symptoms of acute coronary syndrome

More than 95% of participants knew chest pain was a symptom of an acute cardiovascular event (Table 3). However, only 53.5% and 74.8% of participants reported that jaw pain and arm pain are symptoms of an acute event, respectively. Furthermore, 91.9% of participants knew dyspnea was a symptom of an acute event. Most participants reported medical personnel (62.9%) and then the internet (59.9%) as their primary source of information.

## Factors associated with good knowledge among caregivers

Bivariate analysis showed that participants who never went to school or attended regular work were more likely to have poor knowledge (26.4%) compared to participants who were

**Table 1 General demographics of the participants.**

| Variable | Response | Frequency ($n = 469$) | Percentage (%) |
|---|---|---|---|
| Sex | Male | 254 | 54.2 |
| | Female | 215 | 45.8 |
| Living area | Rural | 91 | 19.4 |
| | Urban | 378 | 80.6 |
| Physical activity | Yes | 102 | 21.7 |
| | No | 367 | 78.3 |
| Educational level | None | 10 | 2.1 |
| | Primary education | 29 | 6.2 |
| | Secondary education | 131 | 27.9 |
| | University education | 299 | 63.8 |
| Employment | None | 53 | 11.3 |
| | Student | 161 | 34.3 |
| | Employed | 59 | 12.6 |
| | Free work | 162 | 34.5 |
| | Retired | 34 | 7.2 |
| Income | Less than 500 | 222 | 47.3 |
| | 500–1,000 | 167 | 35.6 |
| | 1,000–1,500 | 68 | 14.5 |
| | 1,500–2,000 | 6 | 1.3 |
| | More than 2,000 | 6 | 1.3 |
| Relationship | Child | 207 | 44.1 |
| | Spouse | 161 | 34.3 |
| | Parent | 34 | 7.2 |
| | Sibling | 49 | 10.4 |
| | Friend | 18 | 3.8 |
| Insured | Yes | 390 | 83.2 |
| | No | 79 | 16.8 |
| General health | Bad | 10 | 2.1 |
| | Moderate | 198 | 42.2 |
| | Good | 261 | 55.7 |
| Health checks | Never | 102 | 21.7 |
| | Within 1 year | 258 | 55.0 |
| | More than 1 year | 109 | 23.2 |
| Family history of cardiovascular diseases | Yes | 138 | 33.9 |
| | No | 269 | 66.1 |
| Family history of early cardiovascular death | Yes | 93 | 20.9 |
| | No | 352 | 79.1 |
| Smoking | Yes | 49 | 10.4 |
| | No | 270 | 57.6 |
| | X-smoker | 150 | 32.0 |

| Table 1 (continued) | | | | |
|---|---|---|---|---|
| **Variable** | **Response** | | **Frequency (*n* = 469)** | **Percentage (%)** |
| Body mass index classification | Underweight | | 8 | 1.7 |
| | Normal | | 149 | 32.2 |
| | Overweight | | 195 | 42.1 |
| | Obese | | 111 | 24.0 |
| Variable | Mean | Standard deviation | Maximum | Minimum |
| Age | 44.38 | 15.92 | 82 | 16 |
| Body mass index | 29.0 | 22.50 | 16.7 | 40.2 |

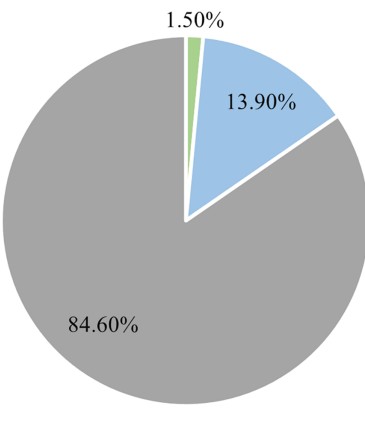

1.50%

13.90%

84.60%

■ Poor <50% ■ Intermediate 50-70% ■ Good ≥70%

**Figure 2 The fund of knowledge among caregivers regarding cardiovascular diseases.**

students, currently working, or retired (*P*-value = 0.011). Participants who were forty years or older had a significantly higher knowledge than their counterparts (*P*-value = 0.039). Moreover, participants with the lowest income category (less than 500 Jordanian dinars) had poor knowledge about CVDs (24.8%) compared to other income categories (*P*-value = 0.0005).

The participant's relationship with the patient was also significantly associated with the level of knowledge; parents or friends had the least amount of knowledge (*P*-value = 0.000). In addition, participants who expressed poor general health status and who never had regular health checks had poor knowledge compared to their healthier counterparts (*P*-value = 0.026, *P*-value = 0.002). Furthermore, participants who did not have CVDs, diabetes or chronic kidney disease had less knowledge compared to their counterparts who had such diseases (*P*-value < 0.050). Additionally, participants who did not have anxiety from CVDs demonstrated significantly lower levels of knowledge (22.5%) in comparison to more anxious participants (*P*-value = 0.000) (Table 4).

**Table 2  Caregivers' knowledge about cardiovascular diseases.**

| Variable | Correct response | Frequency ($n = 469$) | Percentage (%) |
|---|---|---|---|
| Cardiovascular diseases are curable | No | 218 | 46.5% |
| Cardiovascular diseases is main cause of death | Yes | 317 | 67.6% |
| Smoking decrease cardiovascular disease | No | 398 | 84.9% |
| Diabetes increase cardiovascular disease | Yes | 379 | 80.8 |
| Hypertension increase cardiovascular disease | Yes | 448 | 95.5% |
| Excessive alcohol increase cardiovascular disease | Yes | 449 | 95.7% |
| Hyperlipidemia decrease cardiovascular disease | No | 403 | 85.9% |
| Vegetables increase cardiovascular disease | No | 420 | 89.6% |
| Salt intake increase cardiovascular disease | Yes | 425 | 90.6% |
| Obesity increases cardiovascular disease | Yes | 455 | 97.0% |
| Exercise decrease cardiovascular diseases | Yes | 442 | 94.2% |
| Ageing increase cardiovascular diseases | Yes | 389 | 82.9% |
| Men are at higher risk for cardiovascular diseases | Yes | 328 | 69.9% |
| Cardiovascular diseases are not preventable | No | 335 | 71.4% |
| Quitting smoking reduce cardiovascular diseases | Yes | 441 | 94.0% |
| LDL is the bad cholesterol while HDL is the good one | Yes | 334 | 71.2% |
| Blood pressure >140 is hypertension | Yes | 340 | 72.5% |
| Passive smoking is fatal | Yes | 368 | 78.5% |

**Table 3  Caregivers' knowledge about the symptoms of coronary heart disease.**

| Variable | Response | Frequency ($n = 1{,}425$) | Percentage (%) |
|---|---|---|---|
| Chest pain | Yes | 447 | 95.3% |
| Jaw pain | Yes | 251 | 53.5% |
| Shoulder pain | Yes | 351 | 74.8% |
| Dyspnea | Yes | 431 | 91.9% |
| Syncope | Yes | 390 | 83.2% |
| Diplopia | Yes | 268 | 57.1% |
| Source of knowledge | Medical personal | 295 | 62.9% |
| | Friends and family | 249 | 53.1% |
| | Internet | 281 | 59.9% |
| | Television | 105 | 22.4% |
| | Journals | 37 | 7.9% |

**Table 4  Factors associated with good knowledge among caregivers.**

| Variable | | Participants knowledge | | P-value |
|---|---|---|---|---|
| | | <70% $n = 72$ | ≥70% $n = 397$ | |
| Gender | Males $n = 254$ | 12.6% | 87.4% | 0.072 |
| | Females $n = 215$ | 18.6% | 81.4% | |
| Age | <40 ($n = 177$) | 19.8% | 60.2% | 0.039* |
| | >40 ($n = 292$) | 12.7% | 87.3% | |

| Table 4 (continued) | | | | |
|---|---|---|---|---|
| **Variable** | | **Participants knowledge** | | **P-value** |
| | | **<70% *n* = 72** | **≥70% *n* = 397** | |
| Obesity | Yes (*n* = 358) | 15.9% | 84.1% | 0.539 |
| | No (*n* = 111) | 13.5% | 86.5% | |
| Educational level | None *n* = 10 | 20.0% | 80.0% | 0.368 |
| | Primary *n* = 29 | 13.8% | 86.2% | |
| | Secondary *n* = 131 | 19.8% | 80.2% | |
| | University *n* = 299 | 13.4% | 86.6% | |
| Work | None *n* = 53 | 26.4% | 73.6% | 0.011* |
| | Student *n* = 161 | 19.3% | 80.7% | |
| | Employee *n* = 59 | 11.9% | 88.1% | |
| | Free Work *n* = 162 | 11.7% | 88.3% | |
| | Retired *n* = 34 | 2.9% | 97.1% | |
| Income | Less than 500 *n* = 222 | 24.8% | 75.2% | 0.000* |
| | 500–1,000 *n* = 167 | 9.0% | 91.0% | |
| | 1,000–1,500 *n* = 68 | 1.5% | 98.5% | |
| | 1,500–2,000 *n* = 6 | 0.0% | 100.0% | |
| | More than 2,000 *n* = 6 | 16.7% | 83.3% | |
| Residency | Rural *n* = 91 | 17.6% | 82.4% | 0.511 |
| | Urban *n* = 378 | 14.8% | 85.2% | |
| Physical activity | Yes *n* = 102 | 22.5% | 77.5% | 0.023* |
| | No *n* = 367 | 13.4% | 86.6% | |
| Relationship | Child *n* = 207 | 14.0% | 86.0% | 0.000* |
| | Spouse *n* = 161 | 9.3% | 90.7% | |
| | Parent *n* = 34 | 35.3% | 64.7% | |
| | Sibling *n* = 49 | 20.4% | 79.6% | |
| | Friend *n* = 18 | 33.3% | 66.7% | |
| Insurance | Yes *n* = 390 | 22.8% | 77.2% | 0.044* |
| | No *n* = 79 | 13.8% | 86.2% | |
| General health status | Bad *n* = 10 | 20.0% | 80.0% | 0.026* |
| | Moderate *n* = 198 | 10.1% | 89.9% | |
| | Good *n* = 261 | 19.2% | 80.8% | |
| Regular health checks | Never *n* = 102 | 26.5% | 73.5% | 0.002* |
| | Within 1 year *n* = 258 | 13.2% | 86.8% | |
| | More than 1 year *n* = 109 | 10.1% | 89.9% | |
| Cardiovascular diseases | Yes *n* = 171 | 11.1% | 88.9% | 0.049* |
| | No *n* = 298 | 17.8% | 82.2% | |
| Diabetes | Yes *n* = 112 | 4.5% | 95.5% | 0.000* |
| | No *n* = 357 | 18.8% | 81.2% | |
| Chronic kidney disease | Yes *n* = 23 | 0.0% | 100.0% | 0.036* |
| | No *n* = 446 | 16.1% | 83.9% | |

(Continued)

| Table 4 (continued) | | | | |
|---|---|---|---|---|
| **Variable** | | **Participants knowledge** | | **P-value** |
| | | **<70% n = 72** | **≥70% n = 397** | |
| Chronic lung disease | Yes n = 35 | 17.1% | 82.9% | 0.760 |
| | No n = 434 | 15.2% | 84.8% | |
| Stroke | Yes n = 66 | 10.6% | 89.4% | 0.249 |
| | No n = 403 | 16.1% | 83.9% | |
| Family history of cardiovascular diseases | Yes n = 138 | 12.3% | 87.7% | 0.546 |
| | No n = 269 | 14.5% | 85.5% | |
| History of early cardiac death | Yes n = 93 | 9.7% | 90.3% | 0.081 |
| | No n = 352 | 17.0% | 83.0% | |
| Smoking | Yes n = 49 | 4.1% | 95.9% | 0.066 |
| | No n = 270 | 16.3% | 83.7% | |
| | Ex-smoker n = 150 | 17.3% | 82.7% | |
| Anxiety from cardiovascular diseases | None n = 227 | 22.5% | 77.5% | 0.000* |
| | Slight n = 214 | 7.9% | 92.1% | |
| | High n = 28 | 14.3% | 85.7% | |

Note:
* $P$-value < 0.05.

# DISCUSSION

This cross-sectional study aimed to assess caregivers' level of knowledge regarding CVD risk factors and CHD symptomology.

Around 80% of participants in our study exhibited satisfactory knowledge about CVD risk factors, which is consistent with prior research conducted in the UAE (*Kazim et al., 2021*) that had reported a knowledge rate of 71.5%. However, studies from Lebanon (*Machaalani et al., 2022*) (47.3%) and Saudi Arabia (*Mujamammi et al., 2020*) (47.1%) showed lower rates of CVD literacy. These differences in literacy rates could be attributed to the variations in the education levels of study participants and differences in the assessment tools used.

Regarding specific risk behaviors, over 90% of our study participants recognized smoking, hypertension, diabetes, hyperlipidemia, and aging as CVD risk factors. However, there were notable variations in the knowledge rates of individual risk factors reported in regional studies. For example, smoking was recognized as a CVD risk factor by 52–87% of participants, hypertension by 6.2–94%, diabetes by 22–91%, hyperlipidemia by 27–88%, and aging by 50% (*Kazim et al., 2021*; *Mujamammi et al., 2020*; *Maclean, 1999*; *Ansa, Oyo-Ita & Essien, 2007*).

In a local study conducted by *Mukattash et al. (2012)* on 1,000 random Jordanians from the population, knowledge rates for individual risk factors were reported as 75.7%, 6.2%, 5.3%, 4.6%, and 1.5% for smoking, hypertension, diabetes mellitus, hyperlipidemia, and aging, respectively.

In our study, most participants recognized chest pain (95%) and dyspnea (91.9%) as symptoms of an acute cardiovascular event, while knowledge rates for shoulder pain (74.8%) and jaw pain (53.5%) were comparatively lower. The study by *Mukattash et al. (2012)* reported lower knowledge rates for all symptoms, with chest pain recognized by 82% of participants, followed by dyspnea (57%) and shoulder pain (42%). In a study conducted in the United States of America (*Swanoski et al., 2012*), knowledge rates were higher for all symptoms, with chest pain recognized by 92.8% of participants, followed by shoulder pain (86.5%), dyspnea (84.9%), and jaw pain (53.4%). In contrast, a study conducted in Saudi Arabia (*Mujamammi et al., 2020*) reported considerably lower knowledge rates for all symptoms, with chest pain recognized by only 53% of participants, followed by dyspnea (53%), shoulder pain (40%), and jaw pain (28%). These findings suggest there are notable variations in knowledge rates regarding symptoms of acute cardiovascular events among study participants from different regions.

Looking at the factors affecting caregivers' knowledge, older participants had significantly higher knowledge than their counterparts. This finding was consistent with the results of several other studies, where higher age groups had better knowledge about CAD (*Kazim et al., 2021*; *Abukhudair et al., 2022*; *Koohi & Khalili, 2020*; *Awad & Al-Nafisi, 2014*; *Jafary et al., 2005*). Such results might be due to increased medical attention and increased vulnerability to disease that comes with aging, which prompts individuals to access more disease-related information. On the contrary, some studies found that younger age groups had higher knowledge (*Amin, Mostafa & Sarriff, 2014*; *Jamaludin, Jorani & Saidi, 2019*; *Quah et al., 2014*). Hence, effort can be directed towards spreading *via* social media platforms, in hopes of educating more young individuals.

There was a significant association between income and knowledge; more income was linked to a higher level of knowledge. Such results were similar to those of a previous Jordanian study, which concluded that individuals with a lower socioeconomic status had less CVD-related knowledge (*Mukattash et al., 2012*). This is further supported by studies conducted in Lebanon, Uganda, and the United States (*Machaalani et al., 2022*; *Fahs et al., 2017*; *Ndejjo et al., 2020*; *Galbraith et al., 2011*). People with a lower income are less likely to seek health information from healthcare providers and face more challenges interpreting information; hence, having a higher income facilitates health-seeking behaviors (*Tang et al., 2019*). Future campaigns should target their efforts towards lower income communities by utilizing tailored ways of instruction.

Participants who had regular health checkups were more knowledgeable than participants who never had a checkup. Our findings differed from those of *Pallangyo et al. (2020)* who did not find a significant association between frequent health checkups and CVD knowledge amongst caregivers in Tanzania. Routine medical checkups are a type of preventative medicine that can reduce the rates of illness and death at an early stage (*Culica et al., 2002*). Preventative clinic visits, when accompanied by effective communication and action, can promote health literacy, improve patients' understanding of disease, enhance health management, and reduce disease burden (*World Health Organization, 2017*). Enhanced health literacy significantly decreases CVD risk at the primary and secondary levels (*Culica et al., 2002*). Primary care providers should offer

health checkups to caregivers accompanying patients to help control risk factors, promote a healthier lifestyle, and improving their medical knowledge to benefit themselves and the patients they care for. This could further contribute to enhancing caregivers' physical well-being, enabling them to attend to patients without being restricted by attending to their own health needs.

Individuals with a history of diabetes, CVD, or chronic renal disease were more knowledgeable than those without any chronic illnesses. Our findings opposed those of *Pallangyo et al. (2020)*, but other studies conducted in Jordan, Malaysia, and Pakistan reported similar findings (*Mukattash et al., 2012*; *Jafary et al., 2005*; *Amin, Mostafa & Sarriff, 2014*). This indicates that patients with CVD, diabetes, and renal illness are more conscious of their vulnerability to cardiovascular events and, hence, might adopt preventive steps.

Relation to the patient had a significant impact on CHD knowledge, with spouses of patients being the most knowledgeable, followed by children of patients, siblings, parents, and friends, in that order. *Tharu et al. (2022)* reported similar findings in their study, which revealed a significant association between the caregiver-patient relationship and knowledge about pressure injury in spinal cord injuries. This could be attributed to the better communication skills present between married couples, who usually face health-related issues together. One must take into consideration that the spouse is often within the same age group as the patient, thus they have an increased probability for having CVD. Having a history of CVD is a significant factor associated with better knowledge. In our study, 63% of the spouses had a history of CVD, which is statistically significantly higher than other relationship groups.

Participants' perception of their health was a significant factor as well, in which worse perception was associated with poorer knowledge. These results contradicted the findings of *Pallangyo et al. (2020)*, who established a non-significant association. Furthermore, we concluded that the more anxious participants were about their health, the more knowledge they possessed. Such findings can be attributed to anxiety and its role in driving health-seeking behaviors that can enhance one's knowledge (*Asmundson & Taylor, 2020*). Physicians should offer counseling to individuals regardless of whether the patient expressed concerns. They should also identify and advise patients who appear unconcerned regarding their medical condition and are at risk for developing a disease. Furthermore, caregivers may become preoccupied in tending to the needs of the patient, potentially overlooking their own well-being. It is crucial to address this matter, as it has consequences on the overall health of both patients and caregivers.

Despite 97% of the population being aware of the detrimental effects of obesity on CVD, only 32.2% had a normal BMI. Furthermore, 94.2% of the population were aware that physical activity reduces the risk of CVDs, but only 21.7% exercise at least once per week. In addition, smoking is recognized as a risk factor by 84.9% of the population, yet only 57.6% have never smoked. Such responses give the impression that, despite having adequate information, participants are not committed to preventing modifiable risk factors. Our findings were consistent with those of *Koohi & Khalili (2020)* and *Kazim et al. (2021)* who conducted their studies in Iran and the United Arab Emirates, respectively.

Adopting a healthier lifestyle should be one of the top priorities of future campaigns, which should focus on promoting smoking cessation and increasing physical activity. These campaigns may concentrate on actively involving both patients and caregivers, delivering advantages to both parties, and potentially developing increased motivation when collaborative activities are suggested.

Medical personnel were a source of information for 62.9% of the participants. The internet came in second (59.9%), followed by family and friends (53.1%). Television and journals were utilized by 22.4% and 7.9% of our participants, respectively. According to *Hassan, Jarelnape & Elbasheer (2022)*, healthcare staff were the source of information for (88%) of CAD patients in a population in Sudan. Based on a survey done in Nigeria, 64.4% of university workers stated that healthcare staff were their source of knowledge (*Ansa, Oyo-Ita & Essien, 2007*). On the other hand, another study conducted amongst Nigerian teachers and bankers found that television made up 53.8% and 43.8% of the sources of information, while healthcare professionals comprised 7.5% and 13.8%, respectively for teachers and bankers (*Awosan et al., 2013*). These results were somewhat similar to those reported by *Kim et al. (2022)* in their study done amongst Korean women.

The objective of this study was to enhance caregivers' level of knowledge about CVD, particularly in areas where knowledge is lacking. To ensure this occurs, we suggest some recommendations that can be implemented to expand caregiver knowledge, optimize patient care, and promote a lifestyle that can prevent disease. We recommend initiating smoking cessation campaigns as more than a third of our participants have smoked. In addition, promoting an active lifestyle by proposing a few strategies to engage in some physical activity would be of great importance in preventing CVD. New campaigns should target lower-income populations since our study demonstrated that such a population has less knowledge than their wealthier counterparts. Because medical personnel were the primary source of information for most of our participants, physicians should educate both patients and caregivers while simultaneously addressing any concerns that arise during the process. Lastly, proper counseling should be provided to both caregivers who are anxious about developing a CVD and who are unconcerned regarding their medical condition.

This is one of the few studies that explored caregivers' knowledge and attitudes towards CVD risk factors and symptoms. However, several limitations should be acknowledged. First, given that this was a cross-sectional study, the relationship between variables was limited to associations rather than a causal relationship. Second, having a single-center design hinders the generalization of our findings. Additionally, the use of self-administered questionnaires in data collection introduces the possibility of recall bias. Finally, our concluded associations were not adjusted for confounding factors, therefore there is a risk of confounding bias.

## CONCLUSIONS

In conclusion, CVDs are the leading cause of mortality worldwide and fortunately, many are preventable. By gaining adequate knowledge and implementing preventative actions,

one can reduce their chance of developing such diseases. Our study found that most caregivers had good knowledge of CVD risk factors and CHD symptoms. Age, income, frequent health checkups, having a history of CVD, CKD, or DM, and their relationship to the patient were all shown to be associated with the caregivers' level of knowledge. Despite being knowledgeable on CVD risk factors, caregiver responses gave an impression of inadequate adherence to CVD preventive measures. Remarkably, only 71.2% of our population believes CVDs are preventable. The results of this study encourage the development of campaigns, initiatives, and programs that can raise awareness about CVD. Such endeavors, however, need to be developed to target the caregivers' gaps of knowledge while also emphasizing the factors that promote healthier habits. Finally, since healthcare workers are the primary source of information for most patients, they should be encouraged to educate patients and caregivers alike on practical ways by which one can reduce CVD risk.

### Funding
The authors received no funding for this work.

### Competing Interests
The authors declare that they have no competing interests.

### Author Contributions
- Hanna Al-Makhamreh conceived and designed the experiments, performed the experiments, authored or reviewed drafts of the article, and approved the final draft.
- Amro Alkhatib conceived and designed the experiments, performed the experiments, prepared figures and/or tables, authored or reviewed drafts of the article, and approved the final draft.
- Ahmed Attarri conceived and designed the experiments, performed the experiments, authored or reviewed drafts of the article, and approved the final draft.
- Ahmad A. Toubasi conceived and designed the experiments, performed the experiments, analyzed the data, prepared figures and/or tables, authored or reviewed drafts of the article, and approved the final draft.
- Aya Dabbas conceived and designed the experiments, performed the experiments, authored or reviewed drafts of the article, and approved the final draft.
- Basel Al-Bkoor conceived and designed the experiments, performed the experiments, authored or reviewed drafts of the article, and approved the final draft.
- Zaid Sarhan conceived and designed the experiments, performed the experiments, authored or reviewed drafts of the article, and approved the final draft.
- Osama Alghafri conceived and designed the experiments, performed the experiments, authored or reviewed drafts of the article, and approved the final draft.

## Human Ethics

The following information was supplied relating to ethical approvals (*i.e.*, approving body and any reference numbers):

Institutional Review Board, Jordan University Hospital, University of Jordan.

## Data Availability

The data are available in the Supplemental File.

## Supplemental Information

Supplemental information for this article can be found online at http://dx.doi.org/10.7717/peerj.16830#supplemental-information.

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
