# Peer review of "Knowledge of cardiovascular disease risk factors among caregivers of cardiology patients attending Jordan University Hospital"

_PeerJ, doi:10.7717/peerj.16830_

## Round 0.1 · original submission · Major Revisions

Authors need to make changes based on reviewers' comments. Reviewers made strong and useful comments on the manuscript. Thank you.

**Language Note:** The review process has identified that the English language must be improved. PeerJ can provide language editing services - please contact us at [email protected] for pricing (be sure to provide your manuscript number and title). Alternatively, you should make your own arrangements to improve the language quality and provide details in your response letter. – PeerJ Staff

Reviewer 1 ·

Basic reporting

Clear and unambiguous, professional English used throughout: No, see comments below.
Literature references, sufficient field background/context provided: No, see comments below.
Professional article structure, figures, tables. Raw data shared: Acceptable. Data shared.
Self-contained with relevant results to hypotheses: Yes.
a. English grammar needs correction in several places. Please consider an English language editor assist with this.
b. Please use the word “caregiver” instead of the word “caretaker”.
c. The background does not provide sufficient rationale about why the researchers have specifically targeted caregivers for this questionnaire. Why is it necessary for caregivers knowledge of CVD/CHD risk factors to be assessed?
d. Line 125 and 126 – it is unclear what the thresholds are for “good” knowledge/understanding.

Experimental design

Original primary research within Aims and Scope of the journal: Yes.
Research question well defined, relevant & meaningful. It is stated how research fills an identified knowledge gap: This is not clear. What are the implications of measuring the knowledge of CVD/CHD in caregivers?
Rigorous investigation performed to a high technical & ethical standard: Yes, with few exceptions to methods listed below.
Methods described with sufficient detail & information to replicate: Yes.
The research design is appropriate for this study, but there are some points which require clarification:
a. What was the criteria used to determine whether someone was a caregiver? There might have been people who accompany or transport a patient to hospital but do not live with the patient and do not provide care to them.
b. If the patient came in to hospital with more than one caregiver (for instance, if they came in with a son and daughter), were separate questionnaires given to both caregivers? Can the authors report on the number of patients and number of caregivers, is this a 1:1 ratio?

Validity of the findings

Impact and novelty not assessed. Meaningful replication encouraged where rationale & benefit to literature is clearly stated: Yes.
All underlying data have been provided; they are robust, statistically sound, & controlled: Yes.
Conclusions are well stated, linked to original research question & limited to supporting results: See comments below.
There is sufficient comparison with the regional and international literature to explain the findings of this study, however there are no actual recommendations made based on these findings. As one example, if the proportion of smokers is very high in the study population, then it might be worth expanding smoking cessation interventions in the cardiology patient to also include the caregiver. Another example, lower income is associated with lower knowledge on CVD/CHD – does this mean that CVD screening and public health campaigns should be strengthened in lower income communities? For relation, it was found that spouses had better knowledge when compared with other relations – It is possible that the spouses of these patients are also at an age when they have CVD/CHD and thus they already have knowledge on CVD/CHD (through own lived experience) when compared with a son/daughter etc. There are several other paragraphs in the text where the explanations for the findings and how this can specifically be used to tackle CVD/CHD at the primary/secondary prevention level are lacking.

Reviewer 2 ·

Basic reporting

The manuscript was clear however, throughout the article, there were several language errors. I would recomend using an English language professional center to improve the quality of the article.
Literature references were sufficient.

Experimental design

The general structure of the methods section needs to be improved and more detailed to allow replication. Follow the structure provided by the journal. For example instead of having 115 "Consent was taken at the beginning of the survey" there should be a paragraph or subheading for ethics which will emerge the IRB approval, consent, privacy issues during data collection etc. Currently, Line 115 looks like a subheading and mixed up with "instruments"
The "instruments" section could be improved. It should be very clear how many sections were there before they start to decribe them and alittle more detail of what was in each sections. For example line 120, how many questions were there? What score was awarded to each correct question? Line 124-126, What informed the cutoffs? Were the tools used standardized in Jordan? What measures were put in place to make sure that they were standardized? the weighing scale can shift any time?
Data collection is quite unclear, line 123. Was data collected through self administered quetstionnniares or interviews or both? This is unclear. What was the procedure followed so that the study can be replicated. Where were the intereviews carried out? Was there a private room or outside? Who did them? Any training conducted before data collection?
Were there any people who refused to participate in the study and why?

Validity of the findings

"This cross-sectional study aims to assess the level of current knowledge and understanding of CVD risk factors among caretakers of outpatients attending a tertiary-level cardiovascular hospital in Jordan, their awareness of CHD symptoms, and the factors associated with different knowledge levels."
Line 165-166, "The analysis of the overall knowledge of the participants showed that
166 (84.6%) of the participants had more than (75%) of the questions correct (Figure 2)" should come first so that the general picture is known before the specific descriptives of specific answers.
Regarding the factors, why only bivariate analysis? There is need for multivariate analysis to better understand the relationships.

Additional comments

No comment

Reviewer 3 ·

Basic reporting

The authors report a cross-sectional study wherein a self-administered questionnaire was used to assess the knowledge of cardiovascular disease risk factors among caretakers of cardiology patients. Aside from an overly succinct, too concise introduction, this paper is “a keeper.” I believe that a few sentences reviewing each section of the questionnaire used and an explanation of its link to the merit could improve this paper significantly.
References should be up-to-date and following “instructions for authors” (or, in this case – some general citation rules), i.e... when citing a web page, provide a date when it was last accessed.
Starting a table caption with “This table describes…” is too lame for a scientific style (so is the word “lame,”, I know).

Experimental design

-

Validity of the findings

-

Additional comments

-

---

## Round 0.2 · accepted · Accept

The authors incorporated substantial revisions based on feedback from reviewers, enhancing the manuscript's relevance to cardiovascular research. The revised version now aligns with the standards set by the journal. Thanks.

Reviewer 3 ·

Basic reporting

I agree with all the changes.

Experimental design

-

Validity of the findings

-

Additional comments

-